# Interaction of Aggregated Cationic Porphyrins with Human Serum Albumin

**DOI:** 10.3390/ijms24032099

**Published:** 2023-01-20

**Authors:** Mario Samperi, Serena Vittorio, Laura De Luca, Andrea Romeo, Luigi Monsù Scolaro

**Affiliations:** 1CNR—ITAE Istituto di Tecnologie Avanzate per l’Energia “Nicola Giordano”, Via Salita S. Lucia Sopra Contesse 5, 98126 Messina, Italy; 2Dipartimento di Scienze Chimiche, Biologiche, Farmaceutiche ed Ambientali, University of Messina, V. le F. Stagno D’Alcontres, 31, 98166 Messina, Italy; 3CNR—ISMN Istituto per lo Studio dei Materiali Nanostrutturati c/o, Dipartimento di Scienze Chimiche, Biologiche, Farmaceutiche ed Ambientali, University of Messina, V. le F. Stagno D’Alcontres, 31, 98166 Messina, Italy

**Keywords:** porphyrins, aggregates, human serum albumin, chiral supramolecular assemblies

## Abstract

The interaction of an equilibrium mixture of monomeric and aggregated cationic *trans*-5,15-*bis*(*N*-methylpyridinium-4-yl)-10,15-*bis*-diphenylporphine (t-H_2_Pagg) chloride salt with human serum albumin (HSA) has been investigated through UV/Vis absorption, fluorescence emission, circular dichroism and resonant light scattering techniques. The spectroscopic evidence reveals that both the monomeric t-H_2_Pagg and its aggregates bind instantaneously to HSA, leading to the formation of a tight adduct in which the porphyrin is encapsulated within the protein scaffold (S_430_) and to clusters of aggregated porphyrins in electrostatic interaction with the charged biomolecules. These latter species eventually interconvert into the final S_430_ species following pseudo-first-order kinetics. Molecular docking simulations have been performed to get some insights into the nature of the final adduct. Analogously to hemin bound to HSA, the obtained model supports favorable interactions of the porphyrin in the same 1B subdomain of the protein. Hydrophobic and van der Waals energy terms are the main contributions to the calculated ΔG_bind_ value of −117.24 kcal/mol.

## 1. Introduction

Porphyrinoids are an important class of both synthetic and naturally occurring compounds. They possess electronic structures that are responsible for intense absorption bands in the visible region of the spectrum, together with fluorescence emission [1]. Their structural features are enriched by metal ions inserted into the macrocyclic core that bring coordinating properties and redox activity. For these reasons, porphyrins are involved in many biologically relevant molecules where they play important roles in electron transfer (cytochromes), oxygen transport (hemoglobin) and light harvesting (chlorophylls) [2]. Their peculiar optical properties are deeply influenced by the specific microenvironment, making them useful spectroscopic probes [3,4,5,6]. Their propensity to interact with biological systems has also fostered their use as sensitizers for singlet oxygen production upon light irradiation. This ability is at the basis of photodynamic therapy (PDT), where these compounds are actively investigated for producing reactive oxygen species (ROS) able to exert an efficient anticancer activity directly in vivo [7]. 

Porphyrin aggregation is another relevant phenomenon that stems from an interplay of hydrophobic or solvophobic effects together with electrostatics, hydrogen bonding and other specific molecular recognitions [8]. The formation of self-organized or self-assembled supramolecular systems is well documented in the literature and strongly affects the spectroscopic properties [9] and the reactivity [10] with respect to the isolated monomeric species. Furthermore, when porphyrins self-assemble onto chiral templates, chirality could be expressed at the supramolecular level. Indeed, supramolecular chirality plays a fundamental role in nature and in living systems [11,12] and can be exploited in supramolecular chiroptical systems capable of detecting the chiral imprint of molecules and biomolecules [13,14,15,16,17,18].

In this general framework, the interaction between different porphyrin derivatives and proteins have been studied, especially in relation to specific functions. In the case of PDT applications, one of the most investigated one is human serum albumin (HSA) that is the most abundant protein in human blood plasma. It serves as a versatile transporter for numerous endogenous compounds and drug molecules [19,20]. Distinct binding sites were identified by crystallographic studies, most of which comprise positive and negative ionic anchoring units and large hydrophobic pockets [21]. Moreover, the surface-exposed parts of HSA show a high degree of flexibility and qualify the protein to bind a large number of molecular targets [22]. This protein is a single polypeptide chain, arranged in a heart-shaped form owing about 67% α-helix but no β-sheet [23]. HSA possesses three homologous domains, I–III, each of which is comprised of two subdomains, A and B, having common structural elements [24]. Moreover, the isoelectric point of the protein is about 4.8, so that at a neutral pH, this biomolecule is electrically negative, with a net charge on the albumin molecule calculated from its amino acid composition of −15 (mainly due to glutamate and aspartate residues) [24]. The specific binding of hemin in the 1B subdomain of HSA has been reported in the literature [25]. This evidence prompted a series of studies on the design of oxygen transport mimetic systems [26], and to the role of this protein in the transport of many other porphyrins derivatives stabilized in a monomeric form [27]. This latter point is considered rather important in order to obtain an efficient production of singlet oxygen in PDT applications in comparison to aggregated porphyrins [28]. Despite this large number of examples, only a few reports deal with the formation of J-aggregates of the anionic tetrakis-(4-sulfonatophenyl)porphyrin (TPPS_4_) induced by HSA [29]. A couple of examples deal with the stabilization of J-aggregates of TPPS_4_ with chymotrypsin [30] and myoglobin [31]. To the best of our knowledge, no example on the interaction between aggregated cationic porphyrins and HSA has been reported so far.

The cationic *trans*-5,15-*bis*(*N*-methylpyridinium-4-yl)-10,15-*bis*-diphenylporphine (t-H_2_Pagg) and its copper(II) derivative (t-CuPagg) as chloride salts have been proposed in the literature as sensitive chiroptical probes for revealing the conformation and chirality of polypeptides, nucleic acid and bacterium spores [32,33,34,35,36]. These species are dicationic water soluble porphyrins that, upon salt addition, form large fractal assemblies (see Figure 1) whose structural features can be modulated by medium conditions [33,37,38] and even chiral templates [32,39,40]. Fractal clusters of these chromophores display nano- to micrometric sized, flexible and porous structures [41]. The spectroscopic behavior of the two chromophores is rather similar, although the insertion of metal ions into the porphyrin core produces substantial differences. On the one hand, it prevents the protonation of the core, thus suppressing the pH effects on the porphyrin charge under acidic conditions. On the other hand, the metal ion quenches the fluorescence emission (completely in the case of copper(II)), a characteristic well exhibited by the not-metaled form.

On these bases, we thought of interest to investigate the interaction of clusters of t-H_2_Pagg with HSA. Considering the marked similarities between the t-CuPagg porphyrin and its parent-free base form, we have decided to study this latter in order to take advantage of its fluorescence emission properties to obtain more information on the nature of the interaction. Being both the monomeric species and its fractal assemblies positively charged, the initial binding event with the negatively charged HSA should occur mainly by electrostatic interactions. We will show an investigation of the porphyrin–protein system, evaluating the chiroptical and photophysical properties exhibited by the chromophores in the presence of HSA.

## 2. Results and Discussion

As well as t-CuPagg, upon salt addition t-H_2_Pagg self-aggregates in aqueous solution forming large fractal clusters [41], although the amount of salt to obtain the total aggregation of this chromophore is much higher than that necessary for the metal derivative [42].

The extinction spectrum of the neat t-H_2_Pagg porphyrin in phosphate buffer 1 mM pH = 7.4 is shown in Figure 1 (black curve). In line with its monomeric character, it displays a rather sharp Soret band centered at 419 nm together with four weaker Q-bands at a lower energy.

The addition of NaCl (100 mM) promotes the gradual decrease of the B-band and the simultaneous formation of a new band at 450 nm, related to the self-aggregation of porphyrin into fractal assemblies (red curve). The extinction spectra clearly show that this ionic strength value leads to a partial aggregation with an almost equal distribution between monomer and aggregated porphyrin. This experimental finding is in line with previous results and the kinetic profiles displayed in the inset of Figure 1 show a short incubation period, followed by an autocatalytic growth, in agreement with the formation of diffusion limited clusters (DLA), and preceded by a reaction-limited activated step [43].

When a rather high concentration of protein (100 μM) is added to the solution, the extinction spectrum displays an immediate change: the B-band at 419 nm related to the monomeric porphyrin undergoes a bathochromic shift to 430 nm (Figure 2, light green curve), while the spectrum displays an off-set from the baseline probably due to light scattering. This spectrum reveals also that still an almost equal amount of this new species (S_430_) is present together with the porphyrin clusters. During 1 h, the gradual conversion of the band relative to the aggregate into the band at 430 nm occurs, which increases in intensity (dark green curve). This evolution is also reported on the inset of Figure 2, where the intensity at 430 nm recorded over time was fit with Equation (2). The value of *n* obtained from the fit (close to 1) indicates a typical exponential behavior.

The resonant light scattering (RLS) technique confirms similar features observed in extinction (Figure 3). The formation of large supramolecular structures in solution upon salt addition is emphasized by the appearance of an intense signal with an apparent maximum around 470 nm, at the red edge of the absorption band of the clusters (Figure 3, red trace). Upon protein addition, the RLS spectrum shows an instantaneous increase of the signal intensity, followed by a steady reduction. The time-dependence of this signal is reported in the inset of Figure 3, and it shows a similar exponential behavior to that observed previously in the extinction spectra.

Additionally, CD spectroscopy gives important information about the system (Figure 4). Monomeric t-H_2_Pagg, as well as its aggregated form, do not display any CD signals. The addition of protein causes an instantaneous appearance of an exciton-split ICD signal showing negative Cotton effect in the absorption region of the clusters (light green curve), indicating that porphyrin aggregates are responsible for this spectral feature.

These kinds of ICD bands are similar, even if much less intense, to those observed when the t-CuPagg clusters interact with the α-helices of poly(D-glutamic acid) [42], or for t-H_2_Pagg interacting on the surface of M13 bacteriophages [36]. Following over time the spectral evolution, a continuous transformation of the spectrum can be detected, that changes in shape and intensity, leading to a weaker monosignate signal having a negative Cotton effect (dark green curve). These ICD bands are in agreement with the monomeric nature of the porphyrin interacting with the protein [32].

As previously anticipated, fluorescence emission experiments were performed in order to obtain further information. Figure 5 shows the usual fluorescence emission spectrum of monomeric t-H_2_Pagg, characterized by the typical two bands pattern for the aqueous porphyrin solution at 666 and 714 nm (black curve).

As reported in the literature, on increasing the ionic strength, aggregate formation leads to a substantial fluorescence quenching and a slight shift of the maxima, now centered at 673 and 715 nm (red curve) [36].

The addition of HSA causes immediately a relevant modification of the spectrum profile, which becomes much more asymmetric displaying a consistent intensity increase (about five-fold) and a blue shift (≈10 nm) of the main band now centered at 663 nm (light green curve), together with a modest enhancement and a red shift of the band at a longer wavelength (≈9 nm). Both these features slightly increase in intensity over time (dark green curve).

The formation of an adduct between porphyrin and protein is also remarked by the moderate increase of emission quantum yield value of this species ($\phi_{F}$ = 0.072) with respect to that of the free base t-H_2_Pagg, registered in bulk solution ($\phi_{F}$ = 0.06), similar to that reported for the interaction of the same porphyrin with the M13 bacteriophage [36]. This result underlines the assumption of the exclusion of water molecules from the solvation shell of the chromophore as a consequence of the binding to HSA, therefore resulting in effective dielectric changes of the microenvironment around the porphyrins. Thus, it is possible to hypothesize that after the initial electrostatic binding, porphyrins are confined into an internal hydrophobic site of the protein.

This model is also confirmed by the marked difference between the reported static fluorescence anisotropy values for monomeric free porphyrin (r = 0.017) and the porphyrin–protein adduct (r = 0.059). These values suggest a higher rotational freedom for the monomeric species with respect to the adduct form, in line with the binding between the fluorophore and the larger protein [44].

The experimental spectroscopic evidence suggests that the addition of protein on a solution containing a mixture of monomers and clusters shifts instantaneously the equilibrium towards the formation of a new species, S_430_, together with the porphyrin aggregates strongly interacting with HSA. Chirality is immediately induced on the porphyrin assemblies as proved by the bisegnated ICD spectra. As mentioned above, under neutral conditions, the overall charge on the protein is close to −15. Considering that the nominal ratio [HSA]/[t-H_2_Pagg] is 20, a large excess of the protein is present for each porphyrin monomer. Additionally, strong electrostatic interactions between the positively charged porphyrin clusters and the negatively charged HSA should be expected leading to the observed ICD on the aggregates. At the same time, all the spectroscopic features analyzed seem to suggest a preferential interaction between HSA and the monomeric form of t-H_2_Pagg, which leads to the subsequent slow degradation of t-H_2_Pagg aggregates and the formation of an adduct between porphyrin and HSA (S_430_), where the monomeric porphyrin experiences a different and chiral microenvironment (Figure 2).

Molecular Modeling. The interaction between t-H_2_Pagg and HSA was investigated at the molecular level by performing molecular docking simulation employing the crystal structure of HSA in complex with heme (PDB ID 1N5U). Considering that t-H_2_Pagg and heme share similar chemical scaffolds, the docking search was focused on the heme binding site which consist of a hydrophobic D-shaped cavity located in the 1B subdomain of HSA (Figure 6A). The resulting poses were submitted to MM-GBSA calculations, as described in the Methods section, in order to identify the binding conformation with the lowest binding free energy (ΔG_bind_) which afforded a ΔG_bind_ value of −117.24 kcal/mol.

As displayed in Figure 6B, the results revealed that t-H_2_Pagg might occupy the heme binding site on HSA by establishing π-stacking interactions with F134, Y161, F157 and H146. Moreover, several π-cation interactions were observed between (i) one of the pyrrole rings of t-H_2_Pagg and R117 and (ii) one of the N-methylpyridinium moieties and H146, F149 and F157. In addition, one of the pyrrole subunits might engage a H-bond with the hydroxy group of Y161, while electrostatic interactions might be established between E153 and one of the *N*-methylpyridinium rings. Finally, the binding of t-H_2_Pagg to HSA might be stabilized by hydrophobic contacts involving P118, M123, L135, A158, L154, I142 and L115, and van der Waals interactions with V122, A126, V116, F134, F165, L139, L185, S192 and S193.

The contribution of each energy component to the ΔG_bind_, which include electrostatic (ΔG_Coloumb_), van der Waals (ΔG_VdW_), π-π packing (ΔG_Packing_), covalent (ΔG_Covalent_), H-bond (ΔG_H-bond_), self-contacts (ΔG_SelfCont_), hydrophobic (ΔG_Lipo_) and the solvation free energy (ΔG_SolvGB_) [45] are reported in Table 1. According to the outcomes, the most favorable contributions to the ΔG_bind_ derive from the van der Waals and hydrophobic energy terms which therefore represent the primary driving forces for the interaction between t-H_2_Pagg and HSA.

## 3. Materials and Methods

### 3.1. Materials

*trans*-5,15-*bis*(*N*-methylpyridinium-4-yl)-10,15-*bis*-diphenylporphine (t-H_2_Pagg) was purchased from Mid-Century Chemicals as the chloride salt and used as received. Stock solutions of porphyrin were prepared dissolving the solids in dust-free Millipore water and stored in the dark. Solution concentrations were determined from the known molar extinction coefficient at the Soret maximum (ε = 2.40 × 10^5^ M^−1^cm^−1^) [39]. Human serum albumin (HSA) was purchased from Sigma and stock solutions were prepared by dust-free Millipore water in phosphate buffer 0.01 M pH = 7.4. All experiments were carried out in dust-free Millipore water and in 1 mM phosphate buffer, pH = 7.4. All other reagents were supplied by Aldrich Chemicals Co. (St. Louis, MO, USA) and used without further purification.

### 3.2. Methods

UV/Vis measurements were conducted on an Agilent 8453 diode array spectrophotometer. Kinetic experiments were followed in the thermostated compartment of the instrument, with a temperature accuracy of 0.1 K at 298 K. The analyses of the extinction kinetic profiles have been performed by a non-linear fit of the absorption data according to the following equations reported in the literature: (i)
Ext_t_ = Ext_∞_ + (Ext_0_ − Ext_∞_) (1 + (*m* − 1) {*k*_0_ t + (*n* + 1)^−1^ (*k_c_* t)*^n^*^+1^})^−1/(*m*−1)^(1)
where Ext_0_, Ext_∞_, *k*_0_, *k_c_*, *m* and *n* are the parameters to be optimized and, (ii)
Ext_t_ = Ext_0_ + { (Ext_∞_ − Ext_0_) [1 *−* exp(− (*k*t)^n^)]}(2)
where Ext_0_, Ext_∞_, *k* and *n* are the parameters to be optimized (Ext_t_, Ext_0_ and Ext_∞_ are the extinction at time t, at starting time and at the end of aggregation, respectively) [46].

Fluorescence emission and resonance light scattering (RLS) experiments were performed on a Jasco model FP-750 spectrofluorometer equipped with a Hamamatsu R928 photomultiplier, adopting for RLS experiments a synchronous scan protocol with a right-angle geometry [3]. Fluorescence emission was filtered with a high-pass filter (cutoff: 600 nm) to remove excitation overtones. RLS spectra were not corrected for the absorption of the samples. Fluorescence anisotropy measurements were obtained on the same instrument equipped with linear polarizers (Sterling Optics 105UV). Fluorescence anisotropy (r) is defined by the following equation:$$r=\frac{\left(I_{VV}-G\times I_{VH}\right)}{\left(I_{VV}+2G\times I_{VH}\right)}$$
where $I_{VH}$ and $I_{VV}$ are the fluorescence intensities with horizontal and vertical polarization, respectively. The different transmission efficiency of polarized light by both excitation and emission monochromators has been accounted for by correcting the $I_{VH}$ value through the factor $G=I_{HV}/I_{HH}$ that is a correction factor strictly dependent on monochromator wavelength and slit widths [47]. The fluorescence quantum yield ($\phi_{F}$) was calculated on the basis of the following equation:$$\phi_{F}\left(S\right)=\phi_{F}\left(R\right)\times \frac{\left(I_{F}\left(S\right)\right)}{\left(I_{F}\left(R\right)\right)}\times \frac{\left(A_{\lambda ex}\left(R\right)\right)}{\left(A_{\lambda ex}\left(S\right)\right)}$$
where $I_{F}$ is the area under the fluorescence emission spectrum, $A_{\lambda ex}$ is the absorbance value at the excitation wavelength, S refers to the sample, whereas R refers to the reference fluorophore of known quantum yield (tetrakis(*N*-methylpyridinium-4-yl)porphyrin, $\phi_{F}\left(R\right)=$ 0.047 in aqueous buffer solution) [36].

The circular dichroism (CD) spectra were recorded on a JASCO J-720 spectropolarimeter, equipped with a 450 W xenon lamp.

### 3.3. Molecular Modeling

Molecular docking simulation was performed by using the crystal structure of HSA in complex with heme (PDB ID 1N5U) [48]. The protein structure was prepared as described elsewhere [49]. The t-H_2_Pagg structure was built by means of Maestro (Schrödinger Release 2021-4: Maestro, Schrödinger, LLC, New York, NY, USA, 2021) and submitted to a conformational search by means of MacroModel (Schrödinger Release 2021-4: MacroModel, Schrödinger, LLC, New York, NY, USA, 2021) applying the default settings. The charges and geometry of the lowest energy conformation were optimized by QM calculations at B3LYP-D3/LACVP** theory level using Jaguar [50]. The so obtained structure was docked into HSA binding site by means of the software PLANTS v 1.2 [51]. The binding site was set in order to contain all the residues within 10 Å from the co-crystallized ligand. The docking was performed setting speed 1 as accuracy level and ChemPLP as scoring function. The resulting docking poses were rescored by performing MM-GBSA calculation by means of Prime tool [52] employing the VSGB solvation model and “minimize” as sampling method. The minimization involved all the residues situated at 10 Å from the ligand. The pose characterized by the lowest binding free energy value was chosen for the analysis and representation. Ligand–protein interactions were analyzed by means of Discovery Studio Visualizer (BIOVIA, Dassault Systèmes, Discovery Studio Visualizer, v.20.1.0.19295, San Diego, CA, USA: Dassault Systèmes, 2019) and Maestro packages.

## 4. Concluding Remarks

Fractal aggregates of the cationic copper(II) derivative of t-H_2_Pagg are useful chiroptical probes for a variety of biomolecules. These micrometric sized species are positively charged and they readily bind to species bearing negative charges. Moreover, if the substrate is chiral, chirality is transferred to the porphyrin clusters, and usually amplified. In the present investigation, the parent metal-free porphyrin exhibits the same propensity with the additional benefit of being an emitting species in its monomeric form. The interaction of the porphyrin clusters with HSA is a biphasic process: (i) a fast, electrostatically driven contact occurs at the mixing, and chirality is induced on the porphyrin aggregates, and (ii) the fractal aggregates are slowly disrupted by the protein, leading to a quite stable adduct where the porphyrin is bound in the protein scaffold and protected by the solvent. This latter species has been modeled and the porphyrin share the same pocket of hemin in the 1B subdomain of the HSA. Our results show that even cationic porphyrins in aggregated form are able to interact efficiently with serum proteins. Further investigations are on the way to get more insights into the dynamics of this process.

## Data Availability

The data presented in this study are available on request from the corresponding author.

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
