# Peer review of "Interaction of Aggregated Cationic Porphyrins with Human Serum Albumin"

_ijms, 2023, doi:10.3390/ijms24032099_

Round 1

Reviewer 1 Report

The manuscript by Monsù Scolaro et al. reports on the spectroscopic studies of the interaction of aggregates of a dicationic porphyrin derivative with Human Serum Albumin (HSA). The results obtained demonstrate that the macrocycles strongly interact with the HSA by interplay of coulombic and dispersion forces, as indicated by concomitant molecular docking simulations.

The manuscript is well written and the results are interesting and worth of publication. There are just few minor points that need to be clarified before final acceptance.

1) Line 281_The equation 1, does not resemble that cited in the reference 46.

2) Figure 1_A closer look at the spectra reported in Fig.1 shows a not perfect adherence of the theoretical equation to the experimental points. It seems to me that the points follow a “fractal autocatalytic behavior”, with a very shorth lag phase, rather than a RLA (or a stretched exponential with “n” very close to one). My advice is to try to perform a new fit with the widely employed non-conventional equations, maybe reducing the ionic strength. This would certainly decrease the conversion of the monomers into the aggregated forms, but would concomitantly lower the overall aggregation rate, allowing for a better perception of the self-assembly mechanism.

3) Line 130_In caption of Figure 2, replace “chance” by “change”.

Author Response

Dear Editor,

Please find herewith attached a revised manuscript that we modified according to most of the Reviewers comments and suggestions. We take the opportunity to thank both Reviewers for their criticisms that helped us to improve the quality of the manuscript.

Reviewer 1

  • Line 281_The equation 1, does not resemble that cited in the reference 46.

The Reviewer is correct. We are sorry but for some reasons the equation has been wrongly given in the original text. We have corrected the equation according to ref. 46.

  • Figure 1_A closer look at the spectra reported in Fig.1 shows a not perfect adherence of the theoretical equation to the experimental points. It seems to me that the points follow a “fractal autocatalytic behavior”, with a very short lag phase, rather than a RLA (or a stretched exponential with “n” very close to one). My advice is to try to perform a new fit with the widely employed non-conventional equations, maybe reducing the ionic strength. This would certainly decrease the conversion of the monomers into the aggregated forms, but would concomitantly lower the overall aggregation rate, allowing for a better perception of the self-assembly mechanism.

We have performed the fitting procedure on the data reported in the inset of Figure 1, adopting the general equation proposed by Pasternack for the autocatalytic growth of porphyrin assemblies. Actually, the fit is better and we have changed Figure 1 accordingly. We have also introduced a slight modification in the text, commenting the mechanism at the top of p. 4 and we have marked the new equation (now Eq. 1) in the Section 3.2.

  • Line 130_In caption of Figure 2, replace “chance” by “change”.

The typo has been corrected.

Reviewer 2 Report

The manuscript ijms-2149427 entitled “Interaction of Aggregated Cationic Porphyrins with Human Serum Albumin” submitted by L. M. Scolaro et al. describes the interaction of an aggregated dicationic porphyrin derivative with HSA. In my opinion, the manuscript merits to be published in the International Journal of Molecular Science, but some points must be clarified to improve the overall quality of this manuscript.

The porphyrin name “trans-5,15-bis(N-methylpyridinium-4-yl)-10,15-bis-diphenylporphine” as well as the structure in Scheme 1 lacks the counter-ion. Which is the anion used? Iodine? Sulfate? Other? Please, correct both name and structure by adding the counter-ion.

Which could be the influence of the counterion in the formation of the aggregates and their interaction with the human serum albumin (HSA)?

Also, the “N” in the porphyrin name must be in italics.

Page 2, lines 89-91 - The sentence “Different studies reported in the literature have shown the photophysical features of t-H2Pagg when it is interacting with nucleic acids [40] and polypeptides [41], making it a useful photochemical probe to investigate their binding microenvironment.” is redundant. At the beginning of the same paragraph, the authors used different references to do analogous descriptions.

The authors used a dicationic porphyrin bearing positive charges in opposite positions since the formation of the corresponding aggregates was already reported. Do authors consider using the analogous porphyrin bearing positive charges in adjacent meso positions or even the tetra substituted derivative TMePyP? It is possible to prepare aggregates with these porphyrins? In my opinion, the addition of a comparative study comparing the influence of the position of the positive charges and/or the positive charges number will be interesting for this manuscript making it more suitable for IJMS.

Page 6, Line 188 – Please, clarify if is a red or a blue shift. In my opinion, the addition of the HSA does not induce “an abrupt intensity increase” of the two emission bands. The band at 724 nm experienced just a small increase, in fact after 60 min, the intensity of this band is analog to the one displayed by t-H2Pagg before the salt addition. Please, rewrite the discussion of this point to make it clearer for the reader.

Authors claim “the formation of a strongly stabilized adduct between porphyrin and protein…”. Were performed studies to evaluate the reversibility of the binding process to ensure that the porphyrin-HSA complex formation is highly stable? Please calculate stability constants and discuss the value attained with others reported in the literature.

Also, the increase observed in the emission quantum yield after HSA binding does not seem to be relevant and enough to justify the formation of a strongly stabilized adduct. In fact, both the adduct and the free base t-H2Pagg display a very low emission quantum yield (0.072 vs 0.06). Please, improve the discussion of this point.

Which was the compound used as a reference and the solvent (or solvents) used to calculate the emission quantum yield? Add this information in the materials and method section, along with the appropriate literature reference where is reported the emission quantum yield of the compound used as a reference in the solvent used.

Author Response

Dear Editor,

Please find herewith attached a revised manuscript that we modified according to most of the Reviewers comments and suggestions. We take the opportunity to thank both Reviewers for their criticisms that helped us to improve the quality of the manuscript.

Reviewer 2

  • The porphyrin name “trans-5,15-bis(N-methylpyridinium-4-yl)-10,15-bis-diphenylporphine” as well as the structure in Scheme 1 lacks the counter-ion. Which is the anion used? Iodine? Sulfate? Other? Please, correct both name and structure by adding the counter-ion.

In Section 3.1 we already pointed out the chloride as counter-anion of the title porphyrin. Still, in order to make this issue clearer we have remarked the nature of the counter-anion on p. 2 and introduced the description in Scheme 1 and relative caption.

  • Which could be the influence of the counterion in the formation of the aggregates and their interaction with thehuman serum albumin (HSA)?

Actually, the concentration of sodium chloride induces dramatic changes in the final structure of the porphyrin aggregates. Under the experimental conditions used in our investigations, we know that the fractal structure is diffusion-limited DLA type (see e.g. ref. 35, 36, 39). Other studies on the aggregation of water soluble porphyrins (e.g. tetrakis(4-sulfonatophenyl)porphyrin) have shown that counter-anions could exert specific roles (see Occhiuto I. et al .Chem Commun. 2016, 52, 11520-11523). We expect that counter-ions could exert influence on the porphyrin clusters and eventually on their interaction with HSA. We thank the Reviewer for this intriguing question. It surely deserves further investigations that we plan to do in the future.

  • Also, the “N” in the porphyrin name must be in italics.

We have corrected this typo in the text.

  • Page 2, lines 89-91 - The sentence “Different studies reported in the literature have shown the photophysical features of t-H2Pagg when it is interacting with nucleic acids [40] and polypeptides [41], making it a useful photochemical probe to investigate their binding microenvironment.” is redundant. At the beginning of the same paragraph, the authors used different references to do analogous descriptions.

We have deleted the redundant sentence and corrected the references accordingly.

5) The authors used a dicationic porphyrin bearing positive charges in opposite positions since the formation of the corresponding aggregates was already reported. Do authors consider using the analogous porphyrin bearing positive charges in adjacent meso positions or even the tetra substituted derivative TMePyP? It is possible to prepare aggregates with these porphyrins? In my opinion, the addition of a comparative study comparing the influence of the position of the positive charges and/or the positive charges number will be interesting for this manuscript making it more suitable for IJMS.

It is known from the literature (K. Kano et al. J. Phys. Chem. A 1997, 101, 6118) that the tetra-substituted and the tri-substituted derivatives do not aggregate, while the cis-disubstituted species has a tendency to dimerize but not to self-aggregate in large clusters. The aim of the present investigation is on the interaction of fractal clusters of porphyrins with the HSA protein. Therefore, a comparative investigation with these species could be interesting per se, but it is not the main focus of this manuscript.

  • Page 6, Line 188 – Please, clarify if is a red or a blue shift. In my opinion, the addition of the HSA does not induce “an abrupt intensity increase” of the two emission bands. The band at 724 nm experienced just a small increase, in fact after 60 min, the intensity of this band is analog to the one displayed by t-H2Pagg before the salt addition. Please, rewrite the discussion of this point to make it clearer for the reader.

We agree on the fact that the description was not clear. We have changed it accordingly. “The addition of HSA causes immediately a relevant modification of the spectrum profile, which becomes much more asymmetric displaying  a consistent intensity increase (about 5-fold) and a blue shift (≈ 10 nm) of the main band now centered at 663 nm (light green curve), together with a modest enhancement and a red-shift of the band at longer wavelength (≈ 9 nm). Both these features slightly increase in intensity over time (dark green curve).”

  • Authors claim “the formation of a strongly stabilized adduct between porphyrin and protein…”. Were performed studies to evaluate the reversibility of the binding process to ensure that the porphyrin-HSA complex formation is highly stable? Please calculate stability constants and discuss the value attained with others reported in the literature.

Considering the elusive nature of the porphyrin clusters and the difficulty in treating reversibly the porphyrin in the presence of even moderate ionic strength, an evaluation of the binding constant has not been attempted. We agree with the Reviewer that it could be important to obtain such value, but the concomitant aggregation process makes the strategy quite tricky. We surely plan to extend this investigation in the future. Even if far from the experimental value, the binding free energy derived from our molecular modeling could give an idea of the favourable formation of the adduct. We have removed the sentence on the formation of strongly stabilized adduct, thus smoothing the point. At the same time we have slightly remarked the change in the microenvironment as outlined by fluorescence and CD evidences: “At the same time, all the spectroscopic features analyzed seem to suggest a preferential interaction between HSA and the monomeric form of t-H2Pagg, which leads to the subsequent slow degradation of t-H2Pagg aggregates and the formation of an adduct between porphyrin and HSA (S430), where the monomeric porphyrin experiences a different and chiral microenvironment (Scheme 2)”.

  • Also, the increase observed in the emission quantum yield after HSA binding does not seem to be relevant and enough to justify the formation of a strongly stabilized adduct. In fact, both the adduct and the free base t-H2Pagg display a very low emission quantum yield (0.072 vs 0.06). Please, improve the discussion of this point.

We agree on the fact that the change in the fluorescence quantum yield is moderate even if clearly detectable. Anyway, it is in line with the interaction of the t-H2Pagg porphyrin with the surface proteins of various mutants of M13 bacteriophage and the change in the binding environment. The sentence has been smoothed and rewritten accordingly: “The formation of an adduct between porphyrin and protein is also remarked by the moderate increase of emission quantum yield value of this species (QF  = 0.072) with respect to that of the free base t-H2Pagg, registered in bulk solution ( QF = 0.06), similarly to that reported for the interaction of the same porphyrin with M13 bacteriophage [36].”

  • Which was the compound used as a reference and the solvent (or solvents) used to calculate the emission quantum yield? Add this information in the materials and method section, along with the appropriate literature reference where is reported the emission quantum yield of the compound used as a reference in the solvent used.

The fluorescence quantum yields were calculated with respect to tetrakis(N-methylpyridinium-4-yl)porphyrin (0.047 in aqueous buffer solution). The info has been added in the experimental section and a reference has been cited.

Round 2

Reviewer 2 Report

The authors performed significant efforts to improve the manuscript and clarify some points. I believe the manuscript is now suitable for publication in the International Journal of Molecular Sciences.